

# Impact Analysis of processing strategies on Long-term GPS ZTD

Jingna Bai[1], Yidong Lou[1], Weixing Zhang[1], Yaozong Zhou[1], Zhenyi Zhang[1], Chuang Shi[2], Jingnan Liu[1]

[1]GNSS Research Center, Wuhan University, Wuhan, China.
[2]School of Electronic and Information Engineering, Beihang University, Beijing, China.

5    *Correspondence to*:Yidong, Lou (ydlou@whu.edu.cn)

**Abstract:** Homogenized atmospheric water vapor is an important prerequisite for climate analysis. Compared with other techniques, GPS has inherent homogeneity advantage, but it still requires reprocessing and homogenization to eliminate impacts of applied strategy and observation environmental changes where a selection of proper processing strategies is critical. This paper comprehensively investigates an influence of the mapping function, the elevation cut-off angle and homogenization on long-term reprocessing results, in particular for Zenith Tropospheric Delays (ZTD) products, by using GPS observations at 46 IGS stations during 1995 to 2014. In the analysis, for the first time, we included the latest mapping function (VMF3) and exploited homogenized radiosonde data as a reference for ZTD trend evaluations. Our analysis shows that both site position and ZTD solutions achieved the best accuracy when using VMF3 and 3° elevation cut-off angle. Regarding the long-term ZTD trends, results show that the impact of mapping functions is very small, with a maximum difference of 0.3 mm/yr. On the other hand, the discrepancy can reach 2.5 mm/yr by using different elevation cut-off angles. Contrary to recommendations by previous studies, the low elevation cut-off angles (3° or 7°) are suggested for the best estimates of ZTD reprocessing time series when compared to homogenized radiosonde data or ERA5 reference time series. This conclusion has great significance by eliminating the conflict of different optimal elevation cut-off angles for climate analysis and other applications from GNSS data reprocessing.

## 1 Introduction

As the most dominant greenhouse gas, water vapor plays a vital role in the global energy and hydrologic cycle (Kiehl and Trenberth 1997). According to the Clausius–Clapeyron equation, an increase of 1 K in atmospheric temperature will cause an increase of about 7% in water content if the relative humidity is assumed to remain constant, which in turn will further speed up the warming significantly (Trenberth et al. 2003). A strong positive feedback associated with increased water vapor significantly affects the Earth's climate (IPCC 1996). Monitoring variations of atmospheric water vapor is thus important not only for detecting the climate change, but also for a better understanding of a water vapor feedback on the global warming.

In recent years, many studies have been conducted to analyze climatic trends in water vapor time series from radiosonde data, reanalysis data, and Global Positioning System (GPS) data. Radiosonde observations provide the longest water vapor records which have been used to quantify long-term trends (e.g. Ross and Elliott, 2001; Wang et al., 2003; Rowe et al., 2008; Wang et al., 2013; Zhang et al., 2017). However, due to changes in station location, instrument, or operation procedures, radiosonde



data suffer inhomogeneity issue, which leads to ambiguities in long-term trends of water vapor (Dee et al., 2011). Reanalysis data can provide water vapor data with a global coverage, a higher spatial integrity, and a complete record (Lu et al., 2015). However, an inhomogeneity issue in radiosonde humidity data was generally not fixed before being assimilated into reanalysis products (Dee et al., 2011), leading to spurious signals in long-term water vapor trends (Bengtsson et al., 2004; Trenberth et

al., 2005; Qian et al., 2006; Dai et al., 2011). On the other hand, ground-based GPS can provide high-precision, real-time and continuous water vapor distribution information at low cost, and is almost not affected by weather conditions. GPS water vapor is therefore identified as one of the reference data (priority 1) for the Global Climate Observing System (GCOS) Reference Upper-Air Network (GRUAN) (Seidel et al., 2009). Since the early 1990s, GPS has accumulated nearly 30 years of data, but GPS observations have not been assimilated into reanalyses yet, making GPS data an ideal independent observational dataset

that can be used for climate change analyses.

Although GPS has an inherent advantage in homogeneity, updates of GPS data processing models and strategies as well as changes in observing environment can still bring inhomogeneities into the resulting time series. Therefore, to get reliable GPS products for a climate reseach, two procedures are indispensable, namely 1) reprocessing of GPS data by using consistent strategy, such as EPN-Repro1 (Voelksen, 2011) and EPN-Repro2 (Pacione et al., 2017; Dousa et al., 2017), and the three

reprocessing campaigns at IGS (Repro 1/2/3) (Steigenberger et al., 2006; Rebischung et al, 2016; Rebischung et al., 2021), and 2) homogenization of GPS products for eliminating impacts of changes in observing environment.

Many studies have been peformed to study the impact of different strategies on data processing results, in particular the mapping function and the elevation cut-off angle are two factors that were investigated frequently. For example, Vey et al. (2006) and Steigenberger et al. (2009) found that using different mapping functions result in differences in ZTD estimates.

Thomas et al. (2011) confirmed that these differences could be translated into precipitable water vapor (PWV). However, these studies used data for a short period only. For long-term studies, Ning and Elgered (2012) analyzed an influence of eight elevation cut-off angle settings on PWV trends when using 14 years of data at 12 GPS stations in Sweden and Finland. They found that correlations between trends of GPS PWV and radiosonde PWV were the highest when the 25° elevation cut-off angle was used, while the root mean square (RMS) values of PWV errors were the lowest at 10° and 15° elevation cut-off

angles. Baldysz et al. (2018) reprocessed 20 years of data at 20 European stations using eight different strategies. The Precise Point Positioning (PPP) method, GPT2 mapping function and a high elevation cut-off angle were recommended for GPS PWV linear trend estimates.

However, a main deficit in current studies is the utilization of raw radiosonde data or reanalysis products as reference time series, hence introducing additional inhomogeneities to the evaluation of long-term products which makes conclusions

questionable. Dai et al. (2011) proposed a new method for homogenizing radiosonde humidity parameters and generating new homogeneous radiosonde dataset (referred to as Dai dataset hereafter). The Dai dataset has achieved a better homogeneity and a better agreement with GPS long-term trends (Zhao et al., 2015). In this study, we will thus use the Dai dataset as a reference to investigate impacts of different mapping functions and elevation cut-off angles on the long-term GPS tropospheric product. Considering the fact that most GPS stations do not have collocated meteorological measurements for converting ZTD into



PWV, and the use of other resources such as nearby meteorological stations or numerical weather model (NWM) will introduce additional errors to the PWV, this work will pay attention to the ZTD as derived from GPS data processing rather than to the PWV. This is reasonable because the main purpose of this work is to study the impact of data processing strategies on a long-term solution. The latest Vienna Mapping Function, the VMF3 (Landskron and Böhm, 2018), will be included in this kind of analysis for the first time. In addition, the latest global reanalysis, the ERA5, the successor of the widely-used ERA-Interim

(referred to as ERAI hereafter), will be compared and taken as a reference in the homogenization of reprocessing ZTD time series by using a modified penalized maximal $t$ test method (PMTred) (Wang, 2008).

The paper is organized as follows: the datasets, including GPS data, radiosonde data, and reanalysis data (ERA5) and the method to estimate ZTD trends are described in Section 2. Section 3 will focus on analysing the accuracy of estimated coordinates and ZTDs. The method for detecting changepoints is introduced in Section 4 together with assessing impacts of

the mapping function and the elevation cut-off angle settings on estimated ZTD trends. The last section concludes our findings.

## 2 Data and methodology

### 2.1 GPS data

Altogether 46 permanent stations of the IGS network, as displayed in Figure 1, covering the period from 1995 to 2014 were selected in this study, considering two factors: 1) first GPS observations before 1999, and 2) collocated radiosonde stations

within 100 km in horizontal and 100 m in vertical. The GPS observations in 300s intervals were reprocessed on a daily basis using the Position And Navigation Data Analyst (PANDA) software package (Shi et al., 2008) developed at Wuhan University. Precise orbits from the second IGS reprocessing campaign in the IGb08 reference frame were used (Rebischung et al, 2012). Griffith (2019) argued that the IGS repro2 precise clock products are not ideal, suggesting users to either exploit double differences of observations or an explicit clock estimation fully consistent with the orbits. In this study, the satellite clocks

were therefore estimated first by fixing IGS repro2 orbits, and then the ZTD and station positions were estimated in the PPP mode by using the IGS repro2 orbits and the estimated clocks. Altogether 11 experiments, summarized in Table 1, were designed for studying impacts of mapping functions and elevation cut-off angles on reprocessing results. The details of the GPS data processing strategy are given in Table 2.



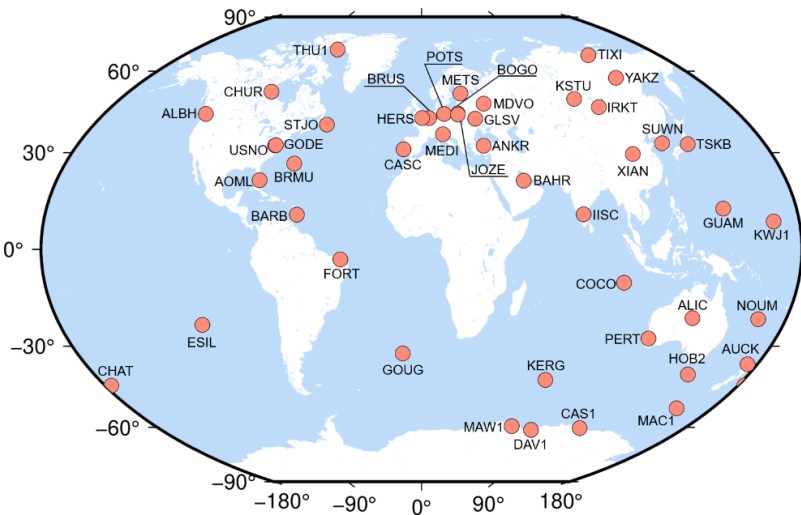

**Figure 1: Geographical distribution of the selected IGS stations.**

**Table 1: Reprocessing solution variants**

| Solution ID | Mapping function | A priori tropospheric delay | Elevation cut-off angle |
|---|---|---|---|
| E1 | GMF | GPT | 7° |
| E2 | GPT2 | GPT2 | 7° |
| E3 | GPT3 | GPT3 | 7° |
| E4 | VMF1 | VMF1 | 7° |
| E5 | VMF3 | VMF3 | 7° |
| E6 | VMF3 | VMF3 | 3° |
| E7 | VMF3 | VMF3 | 10° |
| E8 | VMF3 | VMF3 | 15° |
| E9 | VMF3 | VMF3 | 20° |
| E10 | VMF3 | VMF3 | 25° |
| E11 | VMF3 | VMF3 | 30° |

**Table 2: Data processing strategies for GPS observations**

| | |
|---|---|
| *observation* | |
| Sampling interval | 300 s |
| Frequency combination | Ionosphere-free combination |
| Elevation cut-off angle | See Table1 |
| Elevation weighting strategy | $p=1, e>30°;\ p=2\sin(e),\ e\leqslant 30°$ |
| *Error correction* | |
| Phase center correction | igs08.atx |
| Ocean tide loading | FES2014b |
| A priori tropospheric delay | See Table 1.; |
| Mapping function | See Table 1.; |
| *Parameter estimation* | |
| Satellite orbits | Fixed to IGS repro2 products |
| Satellite clocks | Fixed to estimated 5 min products |
| ZTD stochastic model | Piece-wise constant (1h), random walk between segments ($15mm\sqrt{h}$) |
| Station coordinates | Daily constant |
| Receiver clock corrections | White noise |
| Ambiguities | Fixed |



## 2.2 Reanalysis Data

The ERA5 is the latest atmospheric reanalysis from the European Centre for Medium-Range Weather Forecasts (ECMWF) (Hersbach et al, 2019). It provides a higher resolution and a better performance compared to the ERAI. We will compare GPS ZTDs with ERA5 ZTDs and use the ERA5 ZTD products as a reference to detect changepoints in the GPS ZTD time series. The method for calculating ZTD from ERA5 can be referred to Haase et al. (2003).

## 2.3 Radiosonde data

Altogether 46 radiosonde sites collocated with IGS stations were selected in this study. Dai et al. (2011) detected changepoints first in daily tropospheric dewpoint depression (DPD) time series, then used the most recent segment as a reference to adjust the time series for eliminating discontinuities. Such homogenized time series show generally more realistic long-term trends reported in several previous studies (e.g., Zhao et al, 2012, 2015; Zhang et al, 2019). Therefore, the homogenized radiosonde products processed by Dai et al. (2011) were taken as a reference to evaluate the GPS ZTD trends, while the original radiosonde products (referred to as Raw hereafter) were also compared. The method for calculating the ZTD from radiosonde observation can be referred to Haase et al. (2003).

## 2.4 Trend estimation

ZTD linear trends at GPS stations were estimated using ZTD monthly anomaly time series and the Sen's nonparametric method (Sen, 1968) which has been recognized as more robust than the least-square method (Fan and Yao, 2003). The trend is estimated as

$$trend = median(\frac{x_j - x_i}{t_j - t_i}) \qquad (1)$$

where $x_j$ and $x_i$ are the ZTD monthly anomaly values at time $t_j$ and $t_i$ ($t_j > t_i$). The standard error of the estimated trend is calculated as,

$$s^2 = \frac{\frac{1}{n-2}\sum_1^n (x_i - \hat{x})^2}{\sum_1^n (t_i - \bar{t})^2} \qquad (2)$$

$$\hat{x} = median(x_i) + trend * (t_i - median(t_i)) \qquad (3)$$

where $n$ is a count of data points of the time series, $x_i$ is a ZTD monthly anomaly at time $t_i$, and $\bar{t}$ is the average of all $t_i$.

## 3 Positioning and ZTD error analysis

In this section, the impact of reprocessing strategies on the position and ZTD estimates was studied.





## 3.1 Station position analysis

We use coordinate repeatability to assess the quality of the processing strategies applied in GPS data analysis. Table 3 and 4 summarize the average RMS of coordinate repeatability in the east, north and up components of all stations. Results in Table 3 demonstrate that different mapping functions have small impact on coordinate repeatability, with maximun difference of 0.02, 0.07 and 0.06 mm in the east, north and up component, respectively, and VMF3 shows slightly better results than other mapping functions. On the other hand, by increasing the elevation cut-off angle from 3° to 30°, we can observe obvious increases in coordinate repeatability RMS, especially in the up comopnent, with RMS of 5.87 mm for 3° cut-off angle while

6.64 mm for 30° as shown in Table 4.

**Table 3: Coordinate repeatability by using different mapping functions. (unit: mm)**

| Mapping Function | East RMS | North RMS | Up RMS |
|---|---|---|---|
| GMF | 2.40 | 2.24 | 5.95 |
| GPT2 | 2.39 | 2.17 | 5.92 |
| GPT3 | 2.41 | 2.20 | 5.93 |
| VMF1 | 2.39 | 2.18 | 5.90 |
| VMF3 | 2.39 | 2.17 | 5.89 |

**Table 4: Coordinate repeatability by using different elevation cut-off angles. (unit: mm)**

| Elevation cut-off angle | East RMS | North RMS | Up RMS |
|---|---|---|---|
| 3° | 2.37 | 2.17 | 5.87 |
| 7° | 2.39 | 2.17 | 5.89 |
| 10° | 2.38 | 2.21 | 5.91 |
| 15° | 2.39 | 2.24 | 5.94 |
| 20° | 2.50 | 2.55 | 6.11 |
| 25° | 2.44 | 2.33 | 6.27 |
| 30° | 2.76 | 2.38 | 6.64 |

## 3.2 ZTD error analysis

In addition to the coordinate repeatability, we also assess the ZTD error by taking ERA5-ZTD as reference. We removed the

GPS ZTD outliers by checking ZTD differences (GPS－ERA5) larger than 3 times of a standard deviation of the differences before the comparison, with about 5.5% to 6.3% of the data removed in different experiments. Table 5 and Table 6 present the comparisons by using different mapping functions and different elevation cut-off angles, respectively. We can find a bias of about -0.5 mm in all experiments, indicating a generally larger ZTD from GPS than ERA5. Dousa et al. (2017) and Pacione et al. (2017)  found a bias of about -2 mm in the European GPS reprocessing products from 1996 to 2014 when compared with

ERAI ZTD. Similar to the coordinate repeatility, there are much smaller impacts from mapping functions than from elevation cut-off angles on the ZTD errors. By using VMF3 slightly reduces the ZTD error RMS from 11.02 mm (using GMF) to 10.98 mm. As for different elevation cut-off angle settings, using 3° results in the best performance, with ZTD error RMS of 10.96 mm, compared to 13.73 mm by using 30° cut-off angle, and the 7° cut-off angle setting, which is also commonly used at some analysis centers, has  almost same RMS with 3° setting.



**Table 5: Bias, STD, and RMS of the GPS ZTD and ERA5 ZTD for the different mapping functions. (unit: mm)**

| Mapping Function | Bias | STD | RMS |
|---|---|---|---|
| GMF | -0.57 | 10.41 | 11.02 |
| GPT2 | -0.71 | 10.39 | 11.00 |
| GPT3 | -0.47 | 10.39 | 11.02 |
| VMF1 | -0.62 | 10.36 | 10.99 |
| VMF3 | -0.53 | 10.33 | 10.98 |

**Table 6: Bias, STD, RMS of the GPS ZTD and ERA5 ZTD for the different elevation cut-off angles. (unit: mm)**

| Elevation cut-off angle | Bias | STD | RMS |
|---|---|---|---|
| 3° | -0.54 | 10.31 | 10.96 |
| 7° | -0.53 | 10.33 | 10.98 |
| 10° | -0.53 | 10.39 | 11.06 |
| 15° | -0.44 | 10.58 | 11.34 |
| 20° | -0.59 | 10.94 | 11.80 |
| 25° | -0.28 | 11.53 | 12.52 |
| 30° | -0.59 | 12.66 | 13.73 |

## 4 ZTD Trend analysis

As mentioned above, after the data reprocessing by using consistent models and strategies, the ZTD time series need homogenization for long-term trend analysis, namely, changepoints in GPS ZTD products should be detected and corrected. Impacts of different strategies on long-term trends will be investigated in this section, including the homogenization, mapping function and elevation cut-off angle. Among the 46 IGS stations, 19 stations with GPS and radiosonde common ZTD time series longer than 15 years were selected in the trend analysis.

### 4.1 Changepoint detection and correction

PMTred method proposed by Wang (2008) which can account for a first-order autoregressive noise in time series was used to detect changepoints by checking ZTD monthly differences between GPS and ERA5 datasets. More details can be referred to Ning et al. (2016). We noticed that different strategies can result in different detected changepoints. For example, more changepoints were detected with higher elevation cut-off angle setting, which might be due to the reason that higher elevation cut-off angle could induce larger systematic errors in ZTD time series. As for each changepoint, the offset can be estimated either from the ZTD difference series between GPS and ERA5 (relative correction, denoted as 'REL') or from the GPS ZTD series itself after de-seasonalization (absolute correction, denoted as 'ABS'). An example of the detected changepoints and corresponding estimated offsets using VMF3 and 7° cutoff angle setting is given in Table 7.

**Table 7: Detected changepoints after applying the PMTred test to the monthly mean ZTD difference time series between the GPS (applying VMF3 and 7° settings) and ERA5 data. Bold font means that the changepoint is documented in the log of the GPS sites.**





| Station | Detected date | Offset1(GPS-ERA5)mm | Offset2(GPS) mm | Trend (Before) mm/yr | Trend(REL) mm/yr | Trend(ABS) mm/yr |
|---------|---------------|---------------------|-----------------|----------------------|------------------|------------------|
| ALBH | **199507** 199811 | 2.08 -1.39 | 5.34 0.41 | 0.197 | 0.251 | 0.133 |
| ANKR | 199910 | 7.036 | 2.14 | 0.147 | -0.239 | 0.276 |
| BRMU | 200008 200706 | 3.3213 -2.8906 | 1.38 -2.27 | -0.064 | -0.123 | -0.022 |
| CAS1 | **199912** **201002** **201312** | -4.01 2.50 -2.4 | 2.72 -0.70 -4.58 | 0.043 | 0.145 | -0.041 |
| DAV1 | 200002 **201103** 201303 | -2.19 -2.85 1.27 | 4.62 -2.37 6.64 | 0.088 | 0.306 | -0.204 |
| GODE | 200010 | 3.12 | 5.75 | 0.243 | 0.05 | -0.12 |
| GUAM | 200004 200307 | 8.42 -5.20 | 12.07 1.52 | 0.323 | 0.237 | -0.533 |
| HERS | **199904** **200106** **201008** | -4.70 4.74 3.46 | -2.60 14.93 6.99 | 0.264 | 0.051 | -0.944 |
| HOB2 | **200412** | -1.96 | 0.21 | 0.172 | 0.325 | 0.156 |
| JOZE | 199508 | 4.36 | 6.20 | 0.398 | 0.347 | 0.326 |
| MAC1 | 199611 **200501** **201207** | 6.62 -1.72 2.64 | 7.52 0.51 -0.02 | 0.135 | 0.066 | -0.062 |
| MAW1 | 200002 200602 | -1.78 1.20 | 3.57 1.18 | 0.238 | 0.263 | -0.076 |
| MEDI | 200602 | -3.13 | -3.25 | 0.103 | 0.363 | 0.385 |
| POTS | **200905** **201102** | 2.44 -1.65 | -3.27 4.57 | 0.184 | 0.088 | 0.185 |
| TSKB | 200505 | 2.17 | 2.46 | 0.207 | 0.025 | 0.085 |

160 **4.2 Impact of homogenization method**

Taking ERA5 products as reference, we analysed the GPS ZTD trend after 'REL' correction and 'ABS' correction where an example of GUAM station is illustrated in Figure 2. We can clearly find that all GPS ZTD trends after 'REL' correction are close to ERA5 ZTD trend regardless of processing strategies while this phenomenen is not occoured after 'ABS' correction. Taking the VMF3 and 7° setting as an example, the ZTD trend differences between GPS ZTD and ERA5 products at all

165 stations are displayed in Figure 3. After 'REL' correction, ZTD trend differences at almost all stations are close to zero. We can therefore conclude that the trend after 'REL' correction will be tuned to the trend of the reference product (namely ERA5 in this work). Therefore, in the following sections, only the ZTD time series homogenized by 'ABS' method will be discussed.



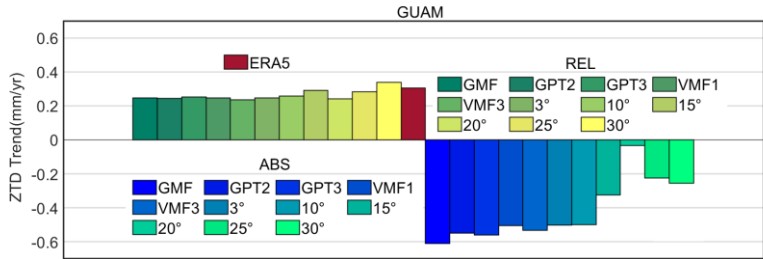

**Figure 2: ERA5 trend (Red bar) and GPS ZTD trend derived from different processing strategies after 'REL' correction (Green bars) and 'ABS' correction (Blue bars) at GUAM station.**

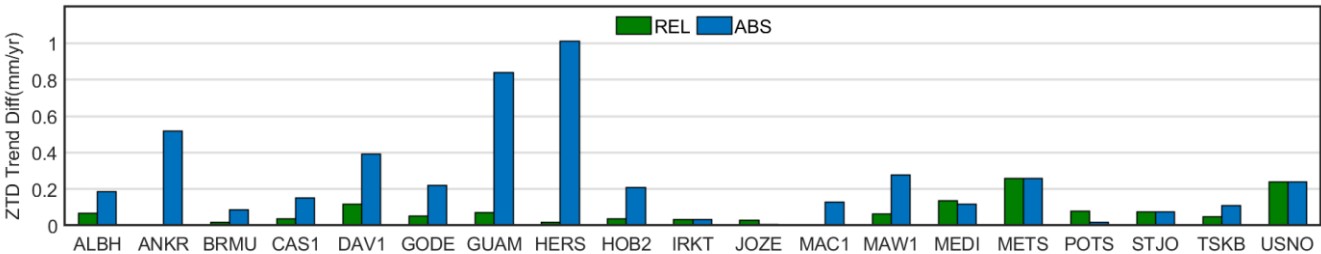

**Figure 3: ZTD trend differences between GPS ZTD after 'REL' correction (Green) and 'ABS' correction' (Red) and ERA5 products at all stations.**

## 4.3 Impact of mapping function

ZTD linear trends estimated from GPS products by applying different mapping functions before (Green Bars) and after 'ABS' (Blue Bars) correction, together with ZTD trends from ERA5 and radiosonde (Raw and Dai) are displayed in Figure 4. We can find that ZTD trends by using different mapping functions before homogenization are close to each other, with differences smaller than 0.3 mm/yr at all stations. Baldysz et al. (2018) also concluded that the PWV products estimated by using different mapping functions showed negligible differences in the trends from 1996 to 2015. However, after homogenization, GPS ZTD trend differences among different mapping functions become larger, such as station BRMU, CAS1 and JOZE. This is mainly due to the reason that different mapping function settings result in different detected changepoints and estimated offsets between segments before and after the changepoint. Figure 5 presents an example of ZTD anlomaly time series derived from using GPT3 and VMF3 at station BRMU where we can find that three changepoints were detected in GPT3 while only two tiny changepoints were found in VMF3 solutions. As a result, trends can be considerably different after homogenization at some stations, indicating a significant impact of homogenization on long-term trend estimation. This impact can also be found for radiosonde data, namely, trends from Dai dataset can show obvious difference from Raw dataset at most stations.



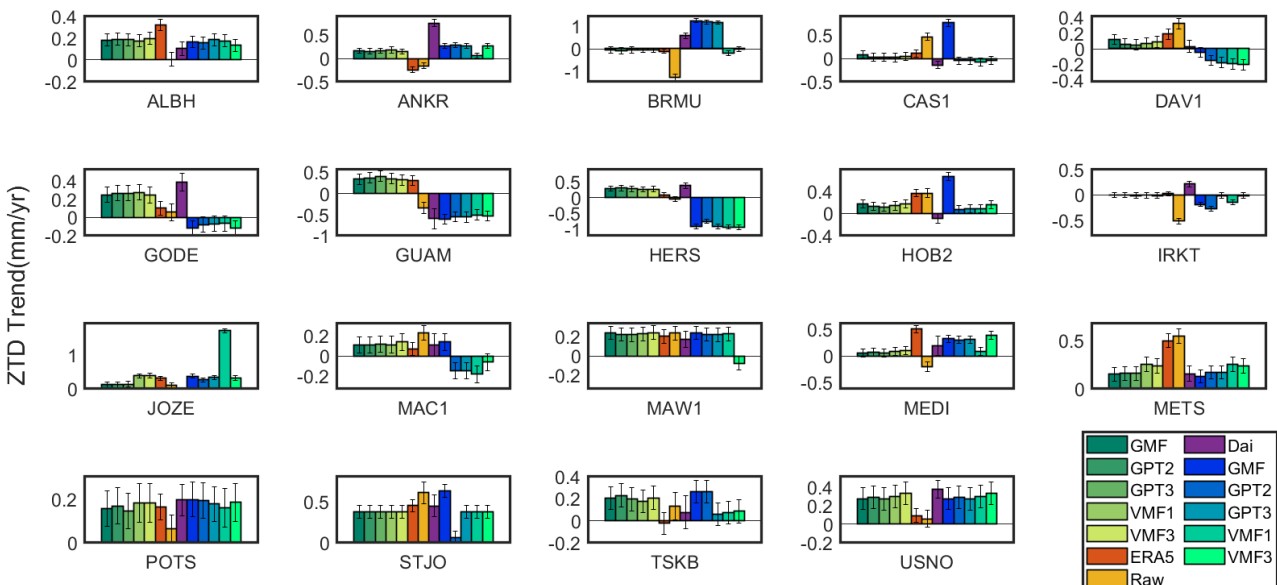

**Figure 4: ZTD linear trends (mm/yr) estimated from GPS products applying different mapping functions before (Green Bars) and after 'ABS' (Blue Bars) correction, ERA5 products and radiosonde data (Raw and Dai products).**

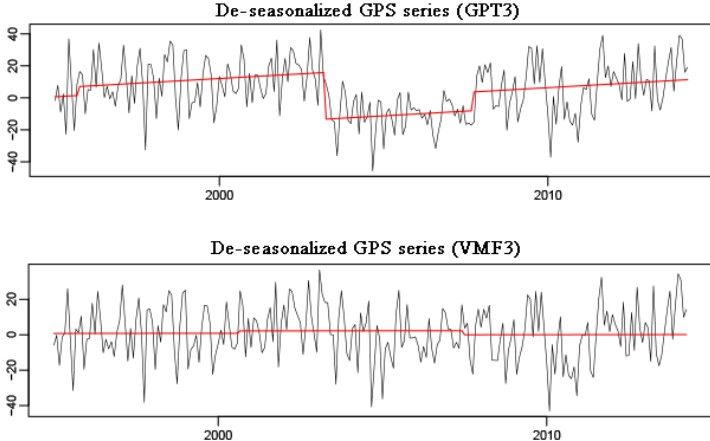

190

**Figure 5: ZTD monthly anomaly time series for GPT3 and VMF3 at station BRMU (unit: mm).**

Taking different data as reference (Dai, Raw, and ERA5), Figure 6 presents the mean value of absolute trend difference for all stations by using different mapping function settings. Again, it can be found that the homogenization enlarges the trend differences among different mapping functions. In addition, the GPS-derived trends generally agree better with Dai than with

195    Raw data, which is consistent with the findings in Zhang et al. (2019). As for different mapping functions, VMF3 shows overall





the best consistency with references, especially with ERA5. However, if taking Dai as reference, GPT3 shows slightly better performance than VMF3.

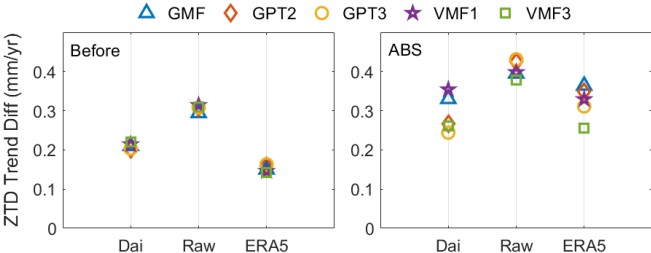

**Figure 6: ZTD trend differences (mm/yr) between GPS (top: before homogenezation; bottom: after homogenization) and different references (Dai, Raw, ERA5) by using different mapping functions**

### 4.4 Impact of elevation cut-off angle

Similarly, Figure 7 presents GPS ZTD trends by using different elevation cut-off angle settings before and after homogenization, together with Dai-, Raw-, and ERA5-derived ZTD trends, and statistics of all stations are shown in Figure 8. It can be noticed that elevation cut-off angles have much larger impact on ZTD trends than mapping function, reaching 2.5 mm/yr at station JOZE between 3° and 30°. This large impact was also reported in previous literatures such as Ning and Elgered (2012) and Dousa et al. (2017). By taking Raw-derived ZTD as reference, GPS ZTD trends before homogenization have the best performance by using high elevation cut-off angle (i.e., 30°), which agrees with conclusions from Ning and Elgered (2012) and Baldysz et al. (2017). On the other hand, when comparing with Dai- and ERA5-derived results, GPS ZTD trends by using low elevation cut-off angles (<15°) show better consistency. As for GPS ZTD time series after homogenization, trends by using 30° elevation cut-off angle have the largest deviations from all refenrences. If taking Dai-derived ZTD trends as reference, the low elevation cut-off angle settings (3° and 7°) have obvious smaller trend errors.



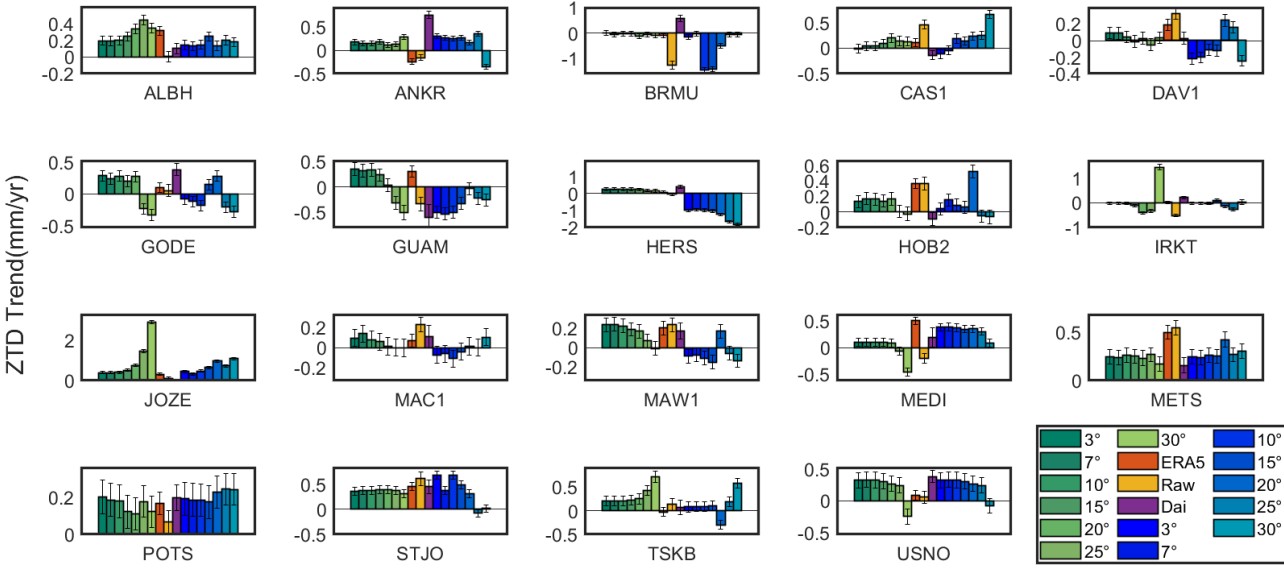

**Figure 7: ZTD linear trends estimated from GPS products applying different elevation cut-off angles before (Green Bars) and after 'ABS' (Blue Bars) correction, ERA5 products and radiosonde data (Raw and Dai products).**

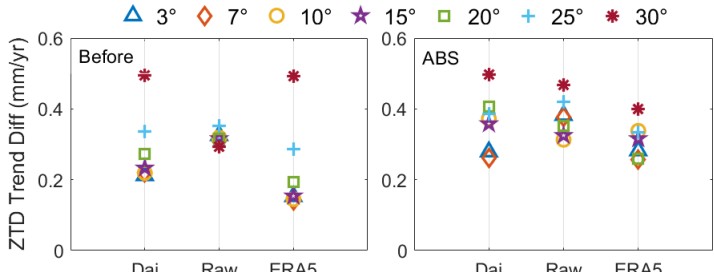

**Figure 8: ZTD trend differences (mm/yr) between GPS (top: before homogenezation; bottom: after homogenization) and different references (Dai, Raw, ERA5) by using different elevation cut-off angle**

## 5 Discussions and Conclusions

Ground-based GPS stations have already accumulated nearly 30-yr of observations since 1990s, which can provide great potential for climate analysis. However, long-term homogeneity is a prerequisite where data reprocessing and homogenization are necessary to eliminate the impact due to changes of data processing strategies and observation environment. This study amis to comprehensively investigate the impact of some key factors in data processing on long-term ZTD trends, including the mapping function, elevation cut-off angle, and homogenization method. By reprocessing GPS data at 46 IGS stations from 1995 to 2014, we firstly evaluated the position repeatability and ZTD error by using different mapping functions and elevation cut-off angle settings. Results show that the elevation cut-off angle setting has much larger impact than mapping function on



the GPS position and ZTD product. Generally, using the latest mapping function, VMF3, gives slightly better solutions than other mapping functions, with position repeatability RMS of 5.89 mm in the up component and ZTD error RMS of 10.98 mm, compared to 5.95 mm and 11.02 mm for using GMF, respectively. As for elevation cut-off angles, both the position and ZTD errors increase with cut-off angles, with best performance achieved by using 3° setting where position repeatability RMS is 2.37, 2.17, 5.87 mm in the east, north and up component, and ZTD error RMS is about 10.96 mm.

We then resorted to the PMTred method to detect and correct changepoints in ZTD time series by using ERA5-derived ZTD as reference. The offset between segments before and after changepoints can be either estimated by using GPS－ERA5 series (relative method) or by using de-seasonalized GPS ZTD itself (absolute method), and we found that the relative method will tune the homogenized ZTD time series to the reference, i.e., ERA5, with almost the same GPS ZTD trends with ERA5 ZTD trends, regardless of the used processing strategies. The impacts of mapping functions, elevation cut-off angles, and homogenization on long-term ZTD trends were then evaluated by comparing with different references, especially, for the first time, including the homogenized radiosonde dataset (Dai) and ERA5. Results show that the homogenization can significantly change the ZTD trends. For using different mapping function, VMF3 shows overall best consistency with references, especially when taking ERA5 as reference. Specifically, if taking Dai as reference, GPT3 shows slightly better performance than VMF3. On the other hand, as for using different elevation cut-off angle settings, GPS ZTD trends before homogenization have the best agreement with Raw radiosonde data by using high elevation cut-off angle (i.e., 30°), which agrees with conclusions from Ning and Elgered (2012) and Baldysz et al. (2017). However, for other situations, i.e., taking Dai- or ERA5-derived ZTD trends as references for un-homogenized GPS ZTD evaluation, or taking any reference for homogenized GPS ZTD evaluation, low elevation cut-off angle settings, especially 3° or 7°, show better performance than high angle settings. This conclusion has great significance for using existed reprocessed GPS products for climate analysis since almost all reprocessing campaigns carried out by GNSS data analysis centers used low elevation cut-off angle setting in their reprocessing works. In other word, tropospheric products generated by these analysis centers can be used for a long-term climate analysis without necessity of reprocessing by using high elevation cut-off angle as suggested by previous studies. One thing we need to notice is that impacts of different factors on the long-term ZTD trends have been discussed in this work, but how large of these impacts compared to the expected trend itself due to the climate change is still absent because it is very hard to exactly know the climate-induced trends. The main reason is that we have few observing techniques that can get reliable ZTD or water vapor trends due to inhomogeneity issue. Some studies have argued that the water vapor content should change with the temperature following the Clausius–Clapeyron equation if the relative humidity is constant (Trenberth et al., 2003). On the other hand, different regions can show quite different scale of CC relationship (Lenderink and Van Meijgaard, 2008). It is therefore not easy to answer how significant are these impacts within the light of the expected trends due to climate change.



**Author contribution**

Yidong Lou and Weixing Zhang proposed the initial ideas. Jingna Bai and Weixing Zhang designed and performed the specific experiments with the help and support of Yaozong Zhou and Zhenyi Zhang. Jingna Bai, Weixing Zhang and Yidong Lou were involved in the manuscript writing. Chuang Shi and Jingnan Liu reviewed this paper and provided suggestions. All authors read and approved the final manuscript.

**Data Availability**

GPS data are provided by IGS which can be accessed from ftps://gdc.cddis.eosdis.nasa.gov/. The reanalysis data, ERA5 and ERA-Interim products, are released by ECMWF at https://www.ecmwf.int/. Radiosonde data are archived at https://www.ncdc.noaa.gov/data-access/weather-balloon/integrated-global-radiosonde-archive/.

**Competing interests**

The authors declare that they have no conflict of interest.

**Acknowledgments**

This work was supported by the National Natural Science Foundation of China (41961144015); Key Research and Development Program of Guangxi Zhuang Autonomous Region, China (2020AB44004); the Fundamental Research Funds for the Central Universities (2042022kf1198). The authors would like to thank IGS for providing GPS data, IGRA for providing radiosonde station observations data and ECMWF for providing ERAI and ERA5 products.

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
