# Peer review of "Impact Analysis of processing strategies on Long-term GPS ZTD"

_EGUsphere, 2023_

## Author Comment (AC1)

**Referee 1:**

**General Comments**

The work presented in the manuscript address the question about long term stability in estimated atmospheric propagation delays using ground-based GNSS stations.

The part I find most interesting, and that may be worth to be published, is the assessment of estimated trends and how these depend on the used mapping functions and the elevation cutoff angle. I think this part is an important contribution to the community but it needs to be more critical. In the present version of the manuscript, I think the results are overrated.

Response: Thank you very much for the recognition of our work and we appreciate all your valuable comments and suggestions. In the revised manuscript, we have addressed all concerns proposed by the reviewer and revised the description of importance of our results, especially in section "Abstract" and "Discussions and Conclusions", which should make our results sound more objective.

We cannot speak about an optimum elevation cutoff angle in general because it is station dependent, i.e. the time dependence of systematic errors in e.g. mapping functions and the multipath environment. Therefore, the presented results are not necessarily in contradiction with those presented by Ning and Elgered (2012) and Baldysz et al. (2018). There is no conflict between these results because the stations analysed in this manuscript have almost no overlap with those in the other studies. Ideally, without systematic errors, the estimated trends shall be identical regardless of the elevation cutoff angle. This is different to the individual ZTD estimates where the geometry obtained for low elevation angles reduce the errors in the estimates (also for the estimated coordinates and especially the vertical). When trends are estimated individual errors are averaged out, if no systematic errors are present.

Response: We also used the same reference data (Raw radiosonde data) as the Ning et al. (2012) and came to a similar conclusion that GPS ZTD trends derived from higher elevation cut-off angle were better. However, if the homogenized RS data is used as a reference, the conclusion is different. Therefore, it can be shown that homogenization has a great impact on the estimated trend. In addition, we agree with you that if there is no systematic bias, the trends by using different elevation cut-off angles are certainly similar. However, in actual data processing, both our results and those

of Ning et al. (2012) and Dousa et al. (2017) show that different elevation cut-off angles may introduce different systematic biases, resulting in non-negligible differences in trends.

*Ning, T., and Elgered, G.: Trends in the atmospheric water vapour content from ground-based GPS: the impact of the elevation cutoff angle. IEEE J-STARS, 5(3), 744–751. https://doi.org/10.1109/JSTARS.2012.2191392, 2012.*

*Dousa, J., Vaclavovic, and P., Elias, M.: Tropospheric products of the second GOP European GNSS reprocessing (1996-2014). Atmos. Meas. Tech., 10, 3589–3607. https://doi.org/10.5194/amt-10-3589-2017, 2017.*

There ought to be a critical discussion about the uncertainties of the estimated trends as a base for a statement regarding which differences that are significant. For example, which differences seen between the estimated trends using the different elevation cutoff angles in Figure 8 are significant. Noting the consequences from introducing changepoints as described, I think this shall be analysed in more detail.

Response: We agree with the reviewer's comment that the significance of the differences and the uncertainties of the trends are closely related. Following your suggestion, we have added the relevant discussion of the trend uncertainties in the revised manuscript. Please see L191-192, L195-197, and L215-216 for the discussion of Figure 4 and Figure 6.

**Specific comments**

Line (L)103: Radiosonde (RS) data were processed by Dai (2011) and are used as a reference. In the study data up to 2014 are used. This requires an explanation. How did you handle RS data acquired in the years thereafter?

Response: The homogenized radiosonde dataset from 1995 to 2012 was provided by Junhong Wang from University at Albany, SUNY. She homogenized the dataset by using method proposed in Dai (2011). We have added this in Acknowledgments, please see L281-282. In our work, for homogenized RS data, only data from 1995 to 2012 were used, and the data after 2012 were absent. We have emphasized this in the revised manuscript. Please see L105-106.

L126: I assume that when mapping functions are compared in Table 3, all these solutions are carried out using an elevation cutoff angle of 7°. Can you mention this explicitly? Please also comment on to what extent you find the differences in Table 3 significant.

Response: Following this suggestion, we have mentioned this explicitly in the captions of Table 3 and Table 4 (now Table 5 and 6 in the revised version). Results in Table 3 (now in Table 5) demonstrate that different mapping functions have small impacts on coordinate repeatability, with maximum difference of 0.02, 0.07 and 0.06 mm in the east, north and up component, respectively. We have no idea about how to estimate uncertainty of the coordinate repeatability, so it is hard to comment on the extent of significance of the differences in Table 3 (now in Table 5). But based on the values of the maximum difference, we can only say that this impact of mapping function is small.

L 140: As I have understood the RMS is defined as the root-sum-squared of the standard deviation and the bias. But this is not the case in Tables 5 and 6. Please explain.

Response: The RMS, STD and bias are average values of all stations. The equation ($RMS^2 = STD^2 + bias^2$) is true for a single station, but not for averages of all stations.

L155: The ABS method suffers from the fact that if an unusual cold and dry month is followed by an unusual warm and humid month a false detection is likely. This ought to be discussed and the different criteria used to identify a changepoint shall be stated.

Response: We used the monthly ZTD time series to detect the changepoints, so the situation you mentioned will only affect one or two points on our monthly ZTD time series. The 'ABS' method used in our study mainly focuses on the shift between segments instead of one or two individual points that deviate the time series, which means this situation should not affect the detection of the 'ABS' method.

L167: I agree with your conclusion that the REL method shall not be used when the goal is to compare "before" and "after" with the ERA5 (because an improvement is expected when the reference data set is used to add changepoints in the GPS time series, the agreement between the trends is of course improved.

Response: Yes, that is the reason we focused on the ZTD time series homogenized by 'ABS' method rather than by 'REL' method.

The robustness of the trend results after adding changepoints can be assessed by studying subsets of the data and the stations.

Response: Following your suggestion, taking the VMF3 and 30° setting as an example, we estimated the GPS ZTD trends for both the full dataset (1995-2014) and subset (2000-2014) before and after homogenization as shown in the Figure below. Only those stations with changepoints detected after 2000 in the GPS ZTD time series are shown. The green and red bars represent trends for full dataset and subset, respectively, with their uncertainties denoted by black bars. It is obvious that the trends of the full dataset and subset agree better after homogenization than those before homogenization, which proves that trends after homogenization are more robust. We also analyzed the trends uncertainties where we can find that the uncertainties are larger for the subsets, illustrating that the data length can affect the uncertainties of the trends.

We are not sure whether we should include these discussions in the manuscript.

[Figure]

**Figure: GPS ZTD trends for both the full dataset (1995-2014, Green bars) and subset (2000-2014, Red bars) before (top) and after (bottom) homogenization and their uncertainties (Black error bars) when using VMF3 and 30° setting at stations with changepoints detected after 2000 in the GPS ZTD time series.**

**Some suggestions related to Table 7:**

**(i) Apply changepoints only for the events that can be supported by the station log.**

Response: In fact, we applied all changepoints in the station log when using PMTred method, but some changepoints were refused. This is due to the fact that not all of the changepoints in the station log can cause significant offsets, which was also found in Ning et al. (2016). In Table 1 of Ning et al. (2016), the changepoints they detected did not include all the changepoints documented in the site log files either.

*Ning T, Wickert J, Deng Z, Heise S, Dick G, Vey S, and Schöne T: Homogenized time series of the atmospheric water vapor content obtained from the GNSS reprocessed data. J Clim 29:2443–2456. https://doi.org/10.1175/JCLI-D-15-0158.1, 2016.*

(ii) Apply only those changepoints when Offset 1 and Offset 2 differ by less than a certain value. The fact that some of them are very different, as well as having opposite signs, I think is warning to be very careful.

Response: Following your comment, we have added this strategy in the revised manuscript. As we all know, the GPS ZTD time series contains some unknown signals, so the ZTD time series used for 'ABS' method only removed seasonal signals and the noise of time series used for 'ABS' is relatively large. It is therefore difficult to set the certain criteria for difference between Offset1 and Offset2. Instead, we only used the changepoints when Offset 1 and Offset 2 have the same sign, namely both positive or negative. We have emphasized this in the revised manuscript. Please see L169-170.

(iii) A combination of (i) and (ii).

Response: It is in fact a combination of (i) and (ii) in the revised manuscript, namely, applying the station log recorded changepoints first, and then following procedure in (ii) to detect additional changepoints. Please see L163-164 and L169-170. In the revised manuscript, all results after homogenization in section 4 are those applying the strategy (iii).

L182: Figure 5: The changepoints seen in the figure are not the ones in Table 7. Are not both of these carried out using an elevation cutoff angle of 7°? Furthermore, the ones in Table 7 are not supported by station logs. I think that if you present such results as in Figure 5 you should discuss

them n more detail and arrive at some understanding why the two mapping functions result in such different trends. Can anyone of them be trusted?

Response: The Table 7 (now Table 10) only showed the changepoints detected in GPS ZTD time series estimated from VMF3 and 7°, not from all solutions using 7°.

Following your comments, we carefully checked our estimated results. We found the strange phenomenon was caused by our mistake. We forgot to exclude the ZTD estimates with the number of GNSS observations being zero. The data processing software will use the a-priori ZTD as output when observations are missing. For different mapping functions, different ZTD priori models were used. For example, GMF used the GPT model, and VMF1 used the ZTD prior values provided by VMF1. This leads to the obvious difference between ZTD from using GPT3 and VMF3 when the observations are missing, resulting in the difference in the detected changepoints.

In the revised manuscript, we have fixed this problem by excluding the ZTD outputs when observations are missing. Now different mapping functions have little influence on the accuracy of the results and the changepoints detection. Please see Table 7 and Figure 4.

Figure 8: Assuming that the work by Dai (2011) implied a significant improvement in the RS data, the results for the Raw comparison may be ignored. Adding that the introduction of changepoints seems to be a rather inaccurate method, the Dai and the ERA5 comparisons before homogenization are the most interesting. It is also worth noting that these two also give the best agreement for elevation cutoff angles of 20° and below. Using GPS satellites only (and not multi-GNSS) means that there are much less observations for cutoff angles above 20°.

Response: We have ignored the Raw comparison in Figure 4 and 6 of revised version, but still retained the Raw results in Figure 5 and 7 for comparison with conclusions from previous studies. For example, taking Raw data as a reference, we came to a similar conclusion to Ning et al. (2012). In the revised manuscript, we also found that Dai and ERA5 give the best agreement for elevation cut-off angles lower than 20°, which is consistent with the fact that the number of GPS satellites is less for cutoff angles above 20°. In addition, the ZTD trends estimated from different elevation cut-off angles are almost the same after homogenization, illustrating that the introduction of changepoints is effective. Please see section 4.4 for more details.

Technical Corrections

Line(L) 6: "Homogenized atmospheric water vapor" sounds strange. To me it sounds like something done in a chemistry lab.?

Response:We have replaced "Homogenized atmospheric water vapor" with "Homogenized atmospheric water vapour data". Please see L6.

L6+: You use the American spelling of vapour, although ACP is a European journal?

Response: We have replaced all 'vapor' into 'vapour' .

L11: the word "latest" may not be true if and when the manuscript is accepted for publication.

Response: We have deleted the word 'latest'.

L14: 0.3 mm/yr   -->   0.3 mm/year   (and a few more places in the manuscript. Note that there is no symbol for "year" in SI, although some use "a", for annual)

Response: We have replaced all 'mm/yr' with 'mm/year'.

L23: 7%   -->   7 %   (see also line 131)

Response: corrected. Please see L22 and L138.

L25: There are more recent IPCC reports. Although it does not change the statement it would be more relevant with a more recent one.

Response: Following your suggestion, we have replaced 'IPCC (1996)' with 'IPCC (2023)'. Please see L24.

*IPCC, 2023: Climate Change 2021. Contribution of Working Group I to the Sixth Assessment Report of the Intergovernmental Panel on Climate Change. https://www.ipcc.ch/report/ar6/wg1/, 2023.*

L80: 300s   --> 300 s

Response: corrected. Please see L79.

Table 1: Perhaps it will be more clear if you note that the E5 solution is used both in the mapping-function comparison and in the elevation cutoff-angle comparison?

Response: Yes, we have split Table 1 into Table 1 and Table 2 and have noted this in the revised manuscript. Please see L86-87.

Table 2: The unit for the random walk shall not be in italic font

Response: corrected. Please see Table 3.

L110: Equation (1) would be informative to explain a bit more so that an overall understanding can be obtained without reading the reference. For example, are the i and j terms all possible combinations (where $t_j > t_i$) or adjacent values only? Please also define "hat x" in Equation (2).

Response: corrected. Please see L116 and L119-120.

Figure 2 (and Figure 5): Remove the text above the graphs and add it with an explanation in the figure captions?

Response: We have removed the text above the graphs in the Figure 2. The Figure 5 is removed.

Figure 4:   Should not the green bars to the right in the graphs be blue (rather than green). The way I interpret the text is that there shall be one green and one blue bar for each mapping function?

Response: Yes, you are right. The text is that there shall be one green and one blue bar (now pink bar in the revised manuscript) for each mapping function. However, we showed the GPS ZTD trends before correction (Green bars) for all mapping function first, then ERA5 and Dai, and finally all corrected GPS ZTD trends (Blue bars).

L189: Y-axis label is missing

Response: We have removed Figure 5.

Figure 6: top and bottom shall read left and right.

Response: corrected. Please see Figure 5, L206 and Figure 7, L228.

L216 (and other places):    homogenezation   -->   homogenization

Response: corrected. Please see L206 and L228.

L219: 30-yr   -->   30 years

Response: corrected. Please see L231.

L285: A doi link is missing, also for some other references and the established standard acronyms for journals are not used in all cases. Furthermore, sometimes they are given as "https://..." addresses and sometimes just as "doi:..."

Response: corrected

---

## Author Comment (AC2)

**Referee 2:**

**General comments:**

The paper "Impact of processing strategies on Long-term GPS ZTD" concerns investigating the influence of the selected GNSS observation processing strategies on the reliability of position and ZTD. In general, the Authors compared the GPS processing approach, which differs in mapping function (GMF, GPT2, GPT3, VMF1, VMF2) and elevation cut-off angle (3°, 7°, 10°,15°, 20°, 25°, 30°). They have mainly focused on the ZTD time series but also provided some basic results regarding position repeatability. Although GNSS meteorology is a well-known concept, it still requires improving existing algorithms and validating possible/new solutions, including new mapping functions. In light of this, the general idea of the paper is justified. However, the complexity of the impact of individual observation processing elements on the reliability of the final solution is very high. Hence it requires a very detailed analysis, which in my opinion, has not been done by the Authors.

Response: We appreciate all the valuable comments and suggestions. All the comments are responded point by point as shown below. We have added more detailed analysis accordingly in the revised manuscript which can hopefully make the reviewer more satisfied.

**Specific comments:**

Firstly, there is no information about trend estimation uncertainties, which are significant when assessing various solutions. Some differences between different observation processing strategies are expected, but assessing their significance is the most important.

Response: In fact, we have given the trend uncertainties in Figure 4 and Figure 7 (now Figure 6), but we did not discuss it accordingly. Following the reviewer's suggestions, we have added analysis about trend estimation uncertainties in the revised manuscript. Please see L195-197 and L215-216.

The Authors have analysed 46 IGS stations, while only 19 have presented results in Figures 4 and 7. There is no appendix to see what is happening with the rest of the stations. Figure 8 presents

results for all stations?, but it is unclear. Additionally – we do not have any information about data quality. The data completeness probably varies for different stations and may affect final solution. ?

Response: 44 IGS stations were selected in this study, considering two factors: 1) having first GPS observations before 1999, and 2) having collocated radiosonde stations within 100 km in horizontal and 150 m in vertical. Among the 44 IGS stations, 19 stations with common ZTD time series between GPS and radiosonde longer than 15 years were selected in the trend analysis. In the revised manuscript, we also excluded months in which GPS observations were less than half the time, so that two more stations (BRMU and JOZE) were rejected. Finally, 17 stations were used for comparison and analysis of trends. We have explained this in L156-158.

Figure 6 (now Figure 5) and Figure 8 (now Figure 7) presents average results for the 19 stations (now 17 in the revised manuscript), we have made it clear in the figure captions in the revised manuscript.

We have added Table 9 to show the data length in the revision. The length of the data is calculated based on the number of the months.

I am also wondering why the Authors have used 1995-2014? Before 2000, quite a poor quality of orbits and SA negatively affect GPS solutions. The station selection is also questionable – 100 km is a lot and may result in different troposphere conditions. Here a table with exact differences between GPS and RS sites is necessary. Moreover, Dai et al. (2011) presented a homogenised dataset until 2011 (or at least that's what the text says). But if the GPS data were processed until 2014, what was the reference for the last three years?

Response: In this study, we reprocessed the GPS ZTD using the IGS repro2 orbits products in order to avoid the inconsistency in the data processing models and strategies. The IGS repro2 covers the period from 1995 to 2014, and we therefore used 1995-2014. SA mainly affects the broadcast ephemeris and should has little effect on the result of reprocessing.

We have added Table 4 to show the distances between GPS and RS stations.

In our study, we only used the homogenized RS data from 1995 to 2012 and the data from 2013 to 2014 were not used. The homogenized radiosonde dataset from 1995 to 2012 was provided by Junhong Wang from University at Albany, SUNY. She homogenized the dataset by using method proposed in Dai (2011). We have added this in the Acknowledgments, please see L281-282.

Why did the Authors decide to verify different mapping functions using 7° cut-off angle?

Response: We need to fix an elevation cut-off angle when comparing different mapping functions. The reason why 7° cut-off angle was used is that some analysis centers (i.e., GFZ, JPL and WHU) also use 7° cut-off angle in data reprocessing, and the position accuracy derived from GMF is slightly worse when using elevation cut-off angles lower than 7° (Dousa et al., 2017 and Qiu et al., 2020).

Dousa, J., Vaclavovic, and P., Elias, M.: Tropospheric products of the second GOP European GNSS reprocessing (1996-2014). Atmos. Meas. Tech., 10, 3589–3607. https://doi.org/10.5194/amt-10-3589-2017, 2017.

Qiu, C.; Wang, X.; Li, Z.; Zhang, S.; Li, H.; Zhang, J.; Yuan, H: The Performance of Different Mapping Functions and Gradient Models in the Determination of Slant Tropospheric Delay. Remote Sens. 2020, 12, 130. https://doi.org/10.3390/rs12010130.

The Authors wrote, "The method for calculating ZTD from ERA5 can be referred to Haase et al. (2003)" – please be more specific about whether used by the Authors method is the same as in Haase, or not. Additionally, what is a temporal and (even more important) spatial (vertical) interpolation between ERA5 and GPS site – there is no information about this.

Response: To make the method clearer, we have replaced "The method for calculating ZTD from ERA5 can be referred to Haase et al. (2003)" with "The method described in Haase et al. (2003) was used for calculating ZTD from ERA5". ERA5 products have improved temporal resolution of 1 h and we also used GPS ZTD products with temporal resolution of 1 h, so there is no need to conduct a temporal interpolation. Regarding the spatial interpolation, we used the method described in Zhang et al. (2017). We have added this information in the revised manuscript. Please see L98-99.

Zhang, W. X., Lou, Y. D., Haase, J., Zhang, R., Zheng, G., Huang, J., et al.: The use of ground-based GPS precipitable water measurements over China to assess radiosonde and ERA-Interim

*moisture rends and errors from 1999 to 2015. Journal of Climate, 30, 7643-7667. https://doi.org/10.1175/JCLI-D-16-0591.1, 2017.*

The ERA5 homogenised dataset (according to Dai et al. 2011) should cover a time span until 2011. Please be more specific about the exact source of radiosonde data (the link given in Dai et al. 2011 does not exist at this moment). This also concerns 'raw Radiosonde'. That would be helpful for the readers. Additionally, please add the info on whether the Authors used exactly the Haase (2003) method for calculating ZTD from RS.

Response: You mean the radiosonde homogenized dataset? The homogenized radiosonde dataset from 1995 to 2012 was provided by Junhong Wang from University at Albany, SUNY. She homogenized the dataset by using method proposed in Dai (2011). We have added this in the Acknowledgments. In the manuscript, we used the homogenized RS data from 1995 to 2012 and the 'raw Radiosonde' products from 1995 to 2014.

To make the method clearer, we have replaced "The method for calculating ZTD from radiosonde observation can be referred to Haase et al. (2003)" with "The method described in Haase et al. (2003) was used for calculating ZTD from radiosonde data". Please see L107-108.

I'm not sure why the Authors have focused on analysing position accuracy since there are no conclusions (just a description of the results) and, more importantly, the results from this part of the manuscript were not considered in any other part. The small variability of position is rather excepted and obtained differences are very small (hundredths of a millimetre).

Response: In this work, we reprocessed the 44 IGS station data from 1995 to 2014 by using different strategies. The main purpose of this work is to access the impact of these strategies on GPS ZTD. Since ZTD and coordinate up component are strongly correlated, we decided to analyze the coordinate accuracy first. Regarding the position accuracy, mapping functions have small impact than cut-off angle, but when the interested product is station position, we do not recommend using cut-off angles higher than 15° in data processing. We have made the conclusions clear in the revised manuscript. Please see L131-132.

There is also no specific conclusion from analysing bias, STD and RMS from differences between GPS ZTD and ERA5 ZTD

Response: Following your suggestion, we have added specific conclusion about GPS ZTD accuracy. Please see L146-147.

There should be more discussion regarding the impact of homogenisation on long-term trends. It is clear that adopting various homogenisation approach influence the final solution the most (since homogenisation may 'fix' even distinct inhomogeneities resulting from adopting various processing strategies). Several papers concern different methods of GNSS time-series homogenisation. It is unclear from what the Authors wrote whether the changepoints they found are correct, better/worse than changepoints that may be found with other approaches.

Response: We agree with the reviewer that different homogenization methods do make a significant difference to trends. The PMTred method has been widely used in long-term trend studies, such as Xu et al. (2013), Ning et al. (2016), and Li et al. (2017). In the revised manuscript, our results also show that the influence of different strategies on ZTD trends is weakened after homogenization, which may illustrate that our homogenization method is effective to some extent. We have added more discussions on the impact of homogenization on long-term trends, please L14, L212-214 and L220-221.

*Xu W., Li Q., Wang X., Yang S., Cao L., and Feng Y.: Homogenization of Chinese daily surface air temperatures and analysis of trends in the extreme temperature indices. Journal of Geophysical Research, 118. https://doi.org/10.1002/jgrd.50791, 2013.*

*Ning T., Wickert J., Deng Z., Heise S., Dick G., Vey S., and Schöne T.: Homogenized time series of the atmospheric water vapor content obtained from the GNSS reprocessed data. J Clim 29:2443–2456. https://doi.org/10.1175/JCLI-D-15-0158.1, 2016.*

*Li Y., Wang G., Han X., Li H., Fan W., Liu K., and Wang H.: Homogenization of Sea Surface Temperature at Xiao Changshan marine station in the east of the Bohai Sea using the PMT method. IOP Conference Series: Earth and Environmental Science, 52, 012055. DOI: 10.1088/1742-6596/52/1/012055, 2017.*

Figures 4 and 5 make me worry about the reliability of the homogenisation process. After taking a closer look at e.g. JOZE station, we can see that the trends are similar before homogenisation, while after homogenisation there is a distinct difference between VMF1 and VMF3. These mapping functions rely on the numerical weather model and are very similar regarding the a and b coefficients. Therefore such differences are unexpected. BRMU station also looks interesting – before homogenisation all trends are similar, after homogenisation, there is a distinct division between climatological and discrete mapping functions. At this point, I would not worry about the comparison to the RS since it may even be 100 km away (there is no info about that).

Figure 5, in turn, makes me worry about the reliability of the GPS observation processing. Presented by the Authors monthly ZTD anomalies present a distinct shift in the case of GPT3 mapping function, while using VMF3 there is no such situation. The main problem is that GPT3 is a climatological mapping function and is therefore continuous. Therefore presented by the Authors shifts in this particular solution are not a problem of GPT3, but of the processing itself.

Response: Following your comments, we carefully checked our estimated results. We found the strange phenomenon was caused by our mistake. We forgot to exclude the ZTD estimates with the number of GNSS observations being zero. The data processing software will use the a-priori ZTD as output when observations missing. For different mapping functions, different ZTD priori models were used. For example, GMF used the GPT model, and VMF1 used the ZTD prior values provided by VMF1. This leads to the obvious difference between ZTD from using GPT3 and VMF3 when the observations are missing, resulting in the difference in the detected changepoints.

In the revised manuscript, we have fixed this problem by excluding the ZTD outputs when observations missing. Now different mapping functions have little influence on the accuracy of the results and the changepoints detection.

I am also not sure why the Authors focus on 'Raw radiosonde' as a reference since they stated in the introduction that RS homogenisation is important. I am also not sure why the Authors focus on the un-homogenized GPS ZTD time-series and, based on them, assess various cut-off angles. "However, for other situations, i.e., taking Dai- or ERA5-derived ZTD trends as references for un-homogenized GPS ZTD evaluation….". If we already know that GPS time series may be affected

by various factors (e.g. antenna/receiver changes), why should we focus on un-homogenized ZTD, while comparing it to the reference set?

Response: We have removed the Raw radiosonde comparison in Figure 4 and 6 in the revised manuscript, but still retained the Raw radiosonde results in Figure 5 and 7 for comparison with conclusions from previous studies, and we found that taking Raw data as a reference, we came to a similar conclusion to Ning et al. (2012).

In the study, we want to analyze not only the imapcts of mapping function and elevation cut-off angle on the long-term GPS ZTD, but also the impact of homogenization on it. Thus, we still retained the un-homogenized GPS ZTD for comparison with the homogenized GPS ZTD to analyze the impact of homogenization, especially the impact of homogenization on the trends of GPS ZTD estimated from different strategies.

To all figures and tables – please change their description to make it possible to correctly understand the presented in them results, without looking for information in the manuscript's main body. Figures are often not well readable.

Response: Following reviewer's suggestions, we have modified all figures and tables to make them well readable. Please see our responses in 'More detailed comments'.

Overall it seems that the presented paper covers too many issues that are too briefly analysed. A proper analysis of each of its elements (i.e. the impact of processing strategy on ZTD, the impact of processing strategy on position and homogenisation on long-term ZTD reliability) is a big task. Therefore it is hard to find reliable outcomes from the conducted analysis.

Response: We agree with the reviewer that the long-term ZTD analysis is a big and complex task and we are trying our best to do some contribution. Specifically, we have added more analyses about ZTD trend uncertainties (L195-197 and L215-216), and have also supplemented each section with clear conclusions based on your suggestions. Please see L131-132 and L146-147.

More detailed comments:

Figure 1  - there is no 'BOGO' station in IGS

Response: Yes, you are right. We checked the list of IGS sites and removed 'BOGO' and 'CASC' stations. Please see Figure 1.

Page 6, Table 4 – please add info that all cut-off angles were tested using VMF3 (I know it was pointed out, but the table should be read correctly, without looking for further information in the manuscript body

Response: corrected. Please see Table 5, L133 and Table 6, L134.

Page 7 Tables 5 and 6 – same as above, but regarding cut-off angle, and mapping function

Response: corrected. Please see Table 7, L148-149 and Table 8, L150-151.

Page 7, Tables 5 and 6 – please add info that this is a difference

Response: We have added this information. Please see Table 7, L148-149 and Table 8, L150-151.

Page 10, Figure 5 – add y-axis description to the figure

Response: We have removed the Figure 5.

Page 9, figure 2 – the colours are way too similar. Instead of the legend, I suggest you add the solution name to the axis

Response: We have add the solution name to the axis. Please see Figure2.

Page 11, line 208 – shouldn't it be Baldysz et al.2016?

Response: It should be Baldysz et al. (2018) and we have corrected it. Please see L218.

Page 12, Figure 8 – The figure description should be corrected (left/right instead of top/bottom)

Response: corrected. Please see Figure 5, L206 and Figure 7, L228.

Page 13, lines 233-235 – this is rather expected. Since we estimate differences between GPS ZTD and ERA5 ZTD and then use these differences to correct GPS ZTD time series, the final GPS ZTD solution will be similar to the ERA5

Response: Yes, you are right. That's why we focused on the ZTD time series homogenized by 'ABS' method rather than by 'REL' method.

---

## Author Response (AR2)

Here, we give the comments from referees, as well as the responses to all comments, and the changes made in the revised manuscript based on the comments.

**1. Referee 1:**

**General**

With interest I have read the responses from the authors to the first round of reviews. I find that all my comments are dealt with. Not all of them are handled as I have suggested, but that is the responsibility of the authors. I still think that the conclusions are somewhat overrated and that more could be done. A main issue is the use of changepoints. It is a good dataset for such studies. For example, IGS stations have high standards and the logs are presumably complete and correct. So why should a changepoint be accepted that is not supported by a logged event?

Response: We appreciate all your valuable comments and suggestions. All the comments are responded point by point as shown below.

The changes that affect the homogeneity of the GPS data can fall into two categories: dataprocessing-related and site-related. The first type of changes is normally due to updates of the reference frame and applied models, different elevation cutoff angles, different mapping functions, and different processing strategies. In this study, the data-processing-related changes have been significantly reduced after a homogenous data reprocessing over the whole time series. The siterelated changes comprise the replacements of hardware (antennas, receivers, cables, and radomes), the differences in the measurements (such as the number of visible GPS satellites and data rate), and the changes in the electromagnetic environment due to, for example, growing vegetation (Pierdicca et al., 2014) and/or different soil moisture (Larson et al., 2010). The changes in the electromagnetic environment can cause different multipath effects on the GPS data, and the resulting errors are normally not fixed in time but varying when reflective properties change and therefore harder to detect and document. The log file for each station only documents hardware changes, accounting for part of the changepoints, which has been reported in many previous literatures. For example, Ning et al. (2016) found that 70 % of the detected changepoints in the GPS-derived IWV time series cannot be related to any documented hardware change after examination 19 years of data at 101 TIGA stations.

Pierdicca, N., Guerriero, L., Giusto, R., Broioni, M., & Egido, A.: SAVERS: A simulator of GNSS reflections from bare and vegetated soils. IEEE Transactions on Geoscience and Remote Sensing, 52, 6542–6554. https://doi.org/10.1109/TGRS.2013.2297572, 2014.

Larson, K. M., Braun, J. J., Small, E. E., Zavorotny, V. U., Gutmann, E. D., & Bilich, A. L.: GPS multipath and its relation to near-surface soil moisture content. IEEE J. Sel. Top. Appl. Earth Obs. Remote Sens., 3, 91–99. https://doi.org/10.1109/JSTARS.2009.2033612, 2010.

Ning T, Wickert J, Deng Z, Heise S, Dick G, Vey S, and Schöne T: Homogenized time series of the atmospheric water vapor content obtained from the GNSS reprocessed data. J Clim 29:2443–2456. https://doi.org/10.1175/JCLI-D-15-0158.1, 2016.

**Specific comments**

I agree with the 2nd reviewer that the short subsection on "station position analysis" is good to remove. The results do not add any new knowledge to what is already well known in the community, and the paper would have a better focus and in agreement with its title.

Response: In this work, we reprocessed the 44 IGS station data from 1995 to 2014 by using different strategies. The main purpose of this work is to access the impact of these strategies on GPS ZTD. Since ZTD and coordinate up component are strongly correlated, we decided to analyze the coordinate accuracy first, and want to illustrate that the same optimal mapping function and cut-off angle setting can be used for both the coordinate and ZTD solutions. In the latest review round, the 2nd reviewer has accepted our response and agreed us to remain this part.

In the previous review I wrote "When trends are estimated individual errors are averaged out, if no systematic errors are present". I missed to add: "Trends are not affected by different systematic errors at for different elevation cutoff angles, as long as they do not also show a time dependence that will alias with the estimated trends." In your response related to this issue you do not reflect upon this issue.

Response: We agree with your comment that the estimated trends would be the same if the systematic errors are time independent. However, differences of solutions by using different cut-off angle settings show time dependence. An example at station GODE is illustrated in the figure below, where we can find differences in both the vertical coordinate and ZTD by using 7° and 30° obviously change with time, which is largely caused by the time-related differences in the number of observations by using different cut-off angle settings.

Figure. (a) The coordinate repeatability time series in up component applying 7° and 30° elevation cut-off angles; (b) The ZTD time series estimated from 7° and 30° solution at GODE station

I think you shall explain how the uncertainties (error bars) in Figure 4 are estimated.

Response: The uncertainty of the estimated trend is calculated as,

$$s^{2} = \frac{\frac{1}{n-2}\sum_{i=1}^{n} (x_{i} - \hat{x})^{2}}{\sum_{i=1}^{n} (t_{i} - \bar{t})^{2}}$$

where *n* is a count of data points of the ZTD time series,  $x_i$  is a ZTD monthly anomaly at time  $t_i$ , and  $\bar{t}$  is the average of all  $t_i$ . The  $\hat{x}$  is estimated from the following equation,

 $\hat{x} = median(x_i) + trend * (t_i - median(t_i))$

Following your suggestion, we have added the explanation. Please see L183-184.

L105+: Using different time periods for the raw (1995-2014) and homogenized (1995-2012) radiosonde data is not ideal when you at a later stage compare these results.

Response: Yes, you are right. In the revised manuscript, we have used the same time period (1995-2012) for different data sets (GPS, ERA5, Raw, and Dai) when estimating ZTD trends. We have added this information in the revised manuscript, please see L167-169. According to the new ZTD results, we modified all the figures and table in the section 4. The conclusions of our study remain the same.

L148: Regarding the relation between Bias, STD, and RMS, I think you shall describe how this is calculated in order to represent many stations. Is bias, STD, and RMS the mean values of those obtained for each station?

Response: Yes, bias, STD, and RMS are the mean values of those obtained for each station. The bias, STD, and RMS are calculated as,

$$Bias = \frac{\sum_{i=1}^{N} \left( \sum_{j=1}^{t} \left( \sum_{j=1}^{T} (ZTD_{Gj} - ZTD_{Ej}) \right) \right)}{N}$$
$$STD = \frac{\sum_{i=1}^{N} \left( \sqrt{\frac{\sum_{j=1}^{t} (dZTD_{j} - \overline{dZTD})^{2}}{t}}{N} \right)}{N}$$
$$dZTD_{j} = ZTD_{Gj} - ZTD_{Ej}$$
$$\overline{dZTD} = \frac{\sum_{i=1}^{t} (ZTD_{Gj} - ZTD_{Ej})}{t}$$
$$RMS = \frac{\sum_{i=1}^{N} \left( \sqrt{\frac{\sum_{j=1}^{t} (ZTD_{Gj} - ZTD_{Ej})^{2}}{t}} \right)}{N}$$

where *N* is the number of the stations. For each station, *t* is the number of ZTD observations in the time series.  $ZTD_{Gj}$  and  $ZTD_{Ej}$  represent the jth GPS and ERA5 ZTD products in the time series, respectively.

Following your comment, we have added the equations for calculating Bias, STD and RMS. Please see Equation (4)-(8).

L 222: I think you mean "smaller trend differences" not "smaller trend errors"?

Response: corrected. Please see L236.

L257-258: You write "The uncertainties are not greatly affected by processing strategies and homogenization method."

I find that surprising because for each additional changepoint introduced, the uncertainty of an estimated linear trend increases. This should be clarified as well as quantified, i.e. how many mm/year correspond to "not greatly"?

Response: We agree with this comment that the uncertainty of an estimated linear trend increases when additional changepoints introduced. We checked the results and found that the uncertainties are reduced after homogenization. We have added the information about uncertainties before and after homogenization in Table 10. Please see Table 10 and L182-183. Meanwhile, we changed "The uncertainties are not greatly affected by processing strategies and homogenization method." to "The homogenization can reduce the uncertainties of the estimated trends." Please see L270-271.

**Technical Corrections**

The unit for hour "h" in Table 3 is still in italic font.

Response: corrected.

L195-196: "... 1 stations have negative trends, ..." --> "... 1 station has a negative trend, ..." Response: corrected. Please see L208.

L251: "... cut-off angle, with the maximum difference reducing from 1.96 to 0.61 mm/year ..." --> "... cut-off angle. The maximum difference was reduced from 1.96 to 0.61 mm/year ..."

Response: corrected. Please see L263-264.

**2. Referee 2:**

I would like to thank the Authors for all their responses to my questions/suggestions. In the present form, the manuscript is much better readable. In the reviewed manuscript, I would also like to correct and clarify several things:

Response: Thank you very much for the recognition of our work and we appreciate all your valuable comments and suggestions. All the comments are responded point by point as shown below.

Page 2, line 46 – should be 'performed'

Response: corrected. Please see L46.

Page 3, line 86 – I suggest "The E5 solution" change to the "The E5 (experiment 5) solution" to show that the shortcuts E\* refers to the different experiments

Response: Following your comment, we have modified the sentence. Please see L87.

In Figure 4, we can clearly see that very often, the ERA5 ZTD and Radiosonde ZTD trends are quite different. Therefore it would be necessary to strongly underline the consistency of GPS solutions.

Response: Following the comment, we have underlined the consistency of GPS solutions in the revised manuscript. Please see L211-212.

A short discussion about GUAM station should be given here since this is the only example when we do have opposite trends from different homogenizations. Although in this case, it is worth underlining that ABS solution follows Radiosonde ones. As in the case of all small islands, considering the size of the land and the model grid box, we can expect some differences. Nevertheless, it should be somehow mentioned. Response: Following your suggestion, we have added a discussion about GUAM station. Please see L210-211.

Page 8, Line 156-157 – now it is clear which stations are analysed and why (thank you for clarifying this). I suggest changing Figure 1 – mark 17 stations that are used for trend analysis with some different colour/sign than stations that are used for position analysis. Or something similar – to distinguish set that is used for trend analysis.;

Response: Following your suggestion, we have modified the Figure 1 in the revised manuscript. 17 stations that are used for trend analysis are marked with green colour in Figure 1.

I also want to clarify one thing because it is still unclear to me. The GNSS ZTD are from 1995 – 2014 (according to the accessibility of the IGS repro2 products). The Radiosonde (homogenized) products are from 1995 – 2012. Does it mean that comparison to ERA5 is for years 1995-2014 (both set), and the comparison to Radiosonde is for years 1995-2012 (both set) or for years 1995-2014 (GNSS) and 1995-2012 (Radiosonde)? Because in the second case the time span between GNSS and Radiosonde is not exactly same, which may cause differences in the trend estimation (according to Baldysz et al. 2016 in AMT, who also analyse the impact of 2 years differences in GNSS time series on the trend estimation).

Response: Yes, you are right. We also read the reference of Baldysz et al. (2016) and found the ZTD time period may affect the trend estimation. Therefore, in the revised manuscript, we have used the same time period (1995-2012) for different data sets (GPS, ERA5, Raw, and Dai) when estimating ZTD trends. We have added this information in the revised manuscript, please see L167-169. According to the new ZTD results, we modified all the figures and table in the section 4. The conclusions of our study remain the same.